# Restoring a Healthy Relationship with Food by Decoupling Stress and Eating: A Translational Review of Nutrition and Mental Health

**DOI:** 10.3390/nu17152466

**Published:** 2025-07-29

**Authors:** Alison Warren, Leigh A. Frame

**Affiliations:** 1Department of Clinical Research and Leadership, The George Washington University School of Medicine and Health Sciences, Washington, DC 20037, USA; leighframe@gwu.ed; 2The Frame-Corr Laboratory, The George Washington University School of Medicine and Health Sciences, Washington, DC 20037, USA; 3The Resiliency & Well-Being Center, The George Washington University, Washington, DC 20037, USA; 4Office of Integrative Medicine and Health, The George Washington University, Washington, DC 20037, USA; 5Department of Physician Assistant Studies, The George Washington University School of Medicine and Health Sciences, Washington, DC 20037, USA

**Keywords:** emotional eating, nutritional psychiatry, gut–brain axis, mindful eating, stress physiology, ultra-processed foods, mental health nutrition, trauma-informed care, precision nutrition

## Abstract

Psychological stress and dietary behavior are interdependent forces that greatly influence mental and physical health. Thus, both what and how we eat impact our well-being. Maladaptive eating patterns, such as eating in response to emotional cues rather than physiological hunger, have become increasingly common amid modern stressors and an ultra-processed food environment. This narrative review synthesizes interdisciplinary findings from nutritional psychiatry, microbiome science, and behavioral nutrition to explore how stress physiology, gut–brain interactions, and dietary quality shape emotional regulation and eating behavior. It highlights mechanisms (e.g., HPA-axis dysregulation, blunted interoception, and inflammatory and epigenetic pathways) and examines the evidence for mindful and intuitive eating; phytochemical-rich, whole-food dietary patterns; and the emerging role of precision nutrition. Trauma-informed approaches, cultural foodways, structural barriers to healthy eating, and clinical implementation strategies (e.g., interprofessional collaboration) are considered in the context of public health equity to support sustainable mental wellness through dietary interventions. Ultimately, restoring a healthy relationship with food positions nutrition not only as sustenance but as a modifiable regulator of affect, cognition, and stress resilience, central to mental and physical well-being.

## 1. Introduction

The inextricable connection between nutrition and brain health has garnered increased recognition underscored by its multidirectional impacts across the lifespan, influencing the epigenome and health outcomes [1]. A salient example is reflected in the reciprocal relationship between eating patterns and mental health outcomes. Indeed, emerging fields such as neuronutrition and nutritional psychiatry are exploring the many ways in which our mental/emotional state affects eating behaviors and, likewise, how we eat and what we eat affect our psychological well-being [2]. Moreover, beverages play a large role alongside foods as part of our daily dietary patterns. While sugar-sweetened beverages have been found to have a deleterious effect on mental health [3], others may exert neuroprotective effects that support mental health. Polyphenol-rich beverages, for instance, such as green tea and coffee have been associated with positive effects on mental health [4]. As part of nutritional neuroscience, neuronutrition explores the influence of diets and dietary components on cognition, behavior, and the prevention and treatment of neuropsychiatric disorders [5].

Further, nutritional psychiatry has illuminated the bidirectional pathways between dietary quality, gut microbiota, inflammation, and mental health [6,7]. Strategies such as intuitive and mindful eating [8,9], phytochemical-rich diets [10], and personalized dietary patterns [11] may hold promise for interrupting maladaptive stress–eating cycles that result in feelings of guilt, disappointment, and negative body image [12].

Emotional eating, a coping mechanism by which individuals over- or underconsume food in response to stress or negative affect, contributes to poor diet quality, obesity, and mental illness [13,14,15]. The American Psychological Association Stress in America survey reports that 38% of adults say they have overeaten or eaten unhealthy foods in the past month because of stress, with 49% reporting these eating behaviors weekly or more, and 30% of adults reported skipping a meal due to stress, with 41% reported doing so weekly or more [12]. As global stress burdens increase, so do rates of disordered eating, particularly among youth and marginalized groups [16,17]. This behavior is reinforced by dysregulation of the hypothalamic–pituitary–adrenal (HPA) axis [13], blunted interoception [18], and the availability of ultra-processed foods (UPFs) that hijack reward pathways [19,20] and are associated with mental health disorders [21,22]. Furthermore, the complexity that arises from the intersectionality of eating behaviors and sex/gender, age, socioeconomic status, geographical location, and cultural norms creates a great deal of complexity and variation that necessitates a precision-focused patient-centered approach. In this paper, “gender” refers to a socially constructed identity that shapes roles, expectations, and behaviors, distinct from biological sex. This distinction is relevant in nutritional and psychological research, as gender influences body image norms, eating behavior, and stress responses in ways not solely attributable to biology (sex) [23]. Importantly, there are major sex and gender differences in stress-related disorders, disproportionately affecting women more than men [24]. This striking disparity may be due, in part, to intersections between stress, inflammation, hormone fluctuations, and epigenetic alterations [25]. Adding to this gap, women are more likely than men to report stress-related unhealthy eating behaviors, 43% and 32%, respectively [12]. This translational narrative review explores the multifaceted relationship between stress and eating behavior, organized around biological mechanisms, eating styles, food quality, public policy, and emergent science.

## 2. Stress, Emotion, and Eating Behavior

The stress response is among the most influential biological triggers of emotional eating behavior [26]. At its core, stress activates the HPA axis, resulting in elevated cortisol secretion, which in turn increases appetite, especially for energy-dense, high-sugar, and high-fat foods [13]. Higher perceived stress has been associated with increased appetite and less restraint in eating in many people but, paradoxically, a decreased intake of food or tendency to skip meals in others [18,26,27]. Despite the well-established pattern of stress-induced eating, it is important to note that the driving mechanisms behind between-person variation are still under exploration, with some research suggesting that females may tend to “comfort eat” more than males [27]. Stress eating that leads to increased consumption is also associated with elevated cortisol levels that may disrupt reward pathways in the brain, particularly within the mesolimbic system, reinforcing compulsive eating as a temporary relief mechanism [13,18]. This neuroendocrine shift creates a physiological feedback loop wherein stress fuels hedonic eating, which then disrupts metabolic regulation and perpetuates emotional dysregulation. Research consistently demonstrates the behavioral implications of this loop, as well as considerable individual variations in regard to food relationships, habitual behaviors, reward, and coping mechanisms [27]. In a qualitative study using thematic analysis, the work of Leow et al. [27], participants reported turning to food during stress not solely due to hunger but as a means of emotional avoidance or numbing. These behaviors often arose in environments where high-calorie foods were accessible, reinforcing stress-induced food choices. Lefebvre et al. [28] differentiated the effects of discrete negative emotional states on cortisol release and sugar preferences, demonstrating that sadness triggered a preference for sugary comfort foods while guilt led to restrictive compensatory behaviors, reflecting a dual pathway that complicates therapeutic interventions. Further research suggests that interoception, which refers to the ability to sense and interpret internal bodily signals and states such as hunger, satiety, and emotional arousal [18], plays a foundational role in both emotional regulation and eating behavior. Blunted interoceptive awareness has been associated with emotional eating, poor satiety recognition, and higher perceived stress [18,29]. Moreover, inflammatory states associated with HPA-axis dysfunction and stress may lead to imprecise (“noisy”) interoception [30], highlighting the multidirectional relationship that stress, mental health, inflammation, and interoception have with eating behaviors.

These biological and psychological pathways are shaped by individual vulnerabilities and regulatory capacities. Evers et al. [31], in a meta-analysis of 56 experimental studies, found a small-to-moderate positive effect of negative mood on food intake, with strong effects observed among individuals with higher emotional eating scores, supporting the notion that trait-based vulnerabilities exacerbate stress–eating cycles. Daily ecological momentary assessment studies [16,32] have added temporal granularity to this model. In adolescents and young adults, greater cortisol reactivity predicted higher caloric intake in response to momentary stress, particularly when paired with impulsive eating traits. Hill et al.’s [33] meta-analysis of 54 studies confirmed that perceived stress reliably predicted increased snack frequency and poorer nutritional choices across age and gender groups. Indeed, emotional eating is documented across all age groups. Hawash et al. [34] highlighted the high prevalence of emotional eating among older adults, noting that, despite increased consumption, older adults may struggle to maintain weight [34]. They also found that age, gender, marital status, higher BMI, and perceived stress are common predictors of emotional eating across a population [34]. Gender-related patterns may be influenced by sociocultural expectations, hormonal cycles, and coping strategies, while age-related changes in metabolism and neuroendocrine regulation further modulate dietary behavior [35]. Ling & Zahry [29] identified emotional eating as a mediator between perceived stress and unhealthy food intake in college students; however, eating self-regulation skills (e.g., planning and interoceptive awareness) attenuated this effect. These findings mirror those from Arias-Magnasco et al. [36], who analyzed data from over 157,000 participants in the UK Biobank, finding that cumulative lifestyle exposures, including sleep, diet, and stress, collectively explained significant variance in depression and anxiety outcomes, underscoring the need to view emotional eating within a broader exposomic framework.

Early life adversity—including adverse childhood experiences (ACEs)—has been strongly associated with dysregulation of the HPA axis and heightened vulnerability to emotional eating in adulthood [37,38]. Chronic exposure to toxic stress during critical developmental windows can sensitize the HPA axis, increasing reactivity to emotional cues and impairing interoceptive awareness, both of which are linked to disordered eating behaviors [39,40]. A recent review by Cascino and Monteleone [41] found that early trauma disrupts the HPA axis by way of neuroendocrine and emotional regulation pathways, predisposing individuals to maladaptive eating behaviors, including binge eating and emotional eating. From a clinical perspective, a recent review of the current clinical evidence linked emotional eating with psychological conditions such as generalized anxiety disorder, major depressive disorder, and stress-related overeating [42]. These associations were particularly strong in populations with low dietary quality and high psychosocial vulnerability. Trauma-informed care is a guiding approach that recognizes the complexity of past trauma (including ACEs), the broad range of impacts on an individual, and the importance of avoiding re-traumatization [43]. Trauma-informed care frameworks emphasize the importance of understanding these underlying patterns, advocating for interventions that not only address maladaptive behaviors (eating behaviors in this case) but also address the long-term neurobiological imprint of early adversity [44]. Integrating trauma-informed principles into nutritional and behavioral interventions (e.g., mindful eating, nutrition self-care) may therefore enhance both engagement and effectiveness in populations with high ACE exposure [45].

Taken together, these findings reinforce that emotional eating is not merely a behavioral quirk but a biologically primed, environmentally reinforced, and emotionally mediated pattern of dysregulation. Its emergence is multifactorial, shaped by neuroendocrine responses, cognitive distortions, emotion regulation deficits, and structural food environments, which may even begin in infancy or childhood [15]. Effective interventions must therefore address both the neurobiology of stress (internal vulnerabilities) and the psychosocial setting (external vulnerabilities) in which emotional eating occurs, targeting the HPA axis alongside the social, behavioral, and dietary context to disrupt the cycle of stress-related eating [15].

## 3. Neurobiology, Gut–Brain Axis, and Inflammation

The microbiota–gut–brain axis (MGBA) has emerged as a foundational concept in nutritional psychiatry. This bidirectional network connects gut microbes, immune pathways, neural circuits, and endocrine systems in regulating behavior, cognition, mood, and emotional states [46]. Diet is the most potent modifiable factor influencing microbial composition and activity, and by extension, neuroinflammation and mood regulation [7,47]. Bremner et al. [6] demonstrated that individuals with high dietary inflammatory indices had elevated systemic inflammation and higher rates of stress, PTSD, and depression. In contrast, short-chain fatty acids (SCFAs), primarily butyrate, acetate, and propionate, are produced when gut microbes ferment dietary fiber; these SCFAs exhibit anti-inflammatory effects and influence brain signaling via the vagus nerve and blood–brain barrier, e.g., permeability [48]. SCFAs are crucial for maintaining gut health; they reduce inflammation, support the mucosal barrier function, regulate gut microbiota, and support electrolyte balance [49]. SCFAs also influence metabolism by activating GPR41 and GPR43 receptors, which stimulate GLP-1 secretion, suppress appetite, and regulate nutrient absorption; disruptions in SCFA production and receptor function have been linked to metabolic disorders such as gestational diabetes mellitus [49]. Aside from appetite and metabolism, alterations in SCFAs have been associated with microbiome disruption, microglial activation, and poor mental health, including serious mental disorders such as schizophrenia [50].

A growing body of evidence supports the role of bioactive components in modifying gut microbiota composition and facilitating health promotion, including supporting mental health [51]. For example, Bordiga & Xu [51] emphasized that polyphenols, terpenoids, and plant-derived bioactive compounds enhance microbial diversity and MGBA functions. In a systematic review, Ribera et al. [52] reported that fermented foods, synbiotics, and probiotics are moderately effective in improving depressive symptoms, especially in individuals with concurrent gastrointestinal distress—likely due to an altered MGBA. Further, Lutz et al. [53] proposed that diet-induced modulation of the MGBA interacts with genetic and epigenetic predispositions (susceptibility) for psychiatric disorders. Notably, individuals with altered BDNF methylation profiles appear to respond more favorably to diets rich in prebiotic fiber and omega-3 fatty acids [53].

Emerging studies continue to refine the microbial signatures associated with psychiatric outcomes. Borrego-Ruiz [54] identified significant microbial signatures of mental disorders, such as reductions in *Faecalibacterium prausnitzii*, that may be modulated through dietary interventions. This aligns with findings by Samuthpongtorn et al. [55], who showed that higher citrus intake increased *F. prausnitzii* abundance and moderated depression risk. Additionally, mental health disorders involving serotonin dysregulation (e.g., anxiety, depression) have been associated with decreased Bifidobacterium and Lactobacillus [50] and have demonstrated symptomatic improvement and cortisol regulation with administration of probiotics that include these strains [50,56]. Collectively, the evidence supports the assertion that mental health outcomes are not only brain-based but gut-modulated. Interventions that support microbiota balance—through whole-food, phytochemical-rich diets—may, thus, represent scalable, non-invasive complements to traditional psychological therapies.

## 4. Eating Styles

### 4.1. Mindfulness and Intuition in Practice

Mindful and intuitive eating are behavioral frameworks that promote internal regulation and a reconnection to interoceptive cues [57]. These approaches directly challenge diet culture’s emphasis on restriction and weight loss by fostering body awareness, emotional acceptance, and unconditional permission to eat when hungry [58]. A recent systematic review by Eaton et al. [57] involving 94,710 individuals found that mindful and intuitive eating were associated with improvements in BMI, diet quality, physical activity, body image, self-compassion, and reduced disordered eating and depressive symptoms. Another systematic review by Grider et al. [8] synthesized 13 studies, finding that mindful and intuitive eating approaches improved dietary quality, reduced emotional eating, and enhanced psychological well-being, with some studies also reporting reductions in BMI. Egan et al. [59] showed that among individuals with cystic fibrosis, a mindfulness practice moderated the association between emotional eating and weight—suggesting that, even in clinical populations, mindfulness may buffer against maladaptive eating behaviors. Similarly, the EATT intervention by Zervos et al. [60], an 8-week mindful eating program in Mediterranean participants with overweight, led to significant reductions in disordered eating scores and improvements in perceived self-regulation, which were maintained at a 3-month follow-up.

Intuitive eating is another positive eating behavior that emphasizes food consumption in accordance with physiological needs, using hunger and satiety cues to direct the quantity, quality, and timing of food consumption [61]. Rochefort et al. [61] extended this framework to biological outcomes, finding that intuitive eating correlated with increased circulating omega-3-derived endocannabinoid mediators (e.g., EPEA and DHEA), which are implicated in mood and inflammation regulation. Goode and Fenton [9] outlined practical implementation strategies for group-based intuitive eating interventions, emphasizing the importance of psychological safety, gradual unlearning of diet rules, and embodiment-based journaling. Knol et al. [62] adapted this work for healthcare providers, who reported improved eating experiences and reduced burnout-related food reliance. Putri et al. [63] noted gender differences in the effectiveness of mindfulness-based interventions on eating behaviors, highlighting the necessity of tailoring delivery formats to accommodate cultural and biological variance. Together, these approaches represent more than tools for weight management; they offer a route to emotional self-regulation, self-trust, and resilience against disordered eating in high-stress contexts.

Interoceptive functioning may also be influenced by dietary patterns, such as the Mediterranean diet, through neurobiological pathways. According to Young et al. [30], components of the Mediterranean diet—such as polyphenols, omega-3 fatty acids, and fermented foods—can reduce neuroinflammation, enhance vagal tone, and support the structural integrity of the insular cortex, a brain region central to interoceptive processing. This reframes nutrition as not only modulating the gut–brain axis as previously discussed, but also strengthening interoceptive brain networks involved in mindfulness, emotion regulation, and behavioral resilience. Integrating dietary strategies that support brain plasticity with interoception-focused behavioral interventions, such as mindful and intuitive eating, may offer a synergistic pathway to reduce stress-related eating and promote sustainable mental health as well as well-being [8,29,30]. Combining health-centric dietary patterns with health-centric dietary behaviors promotes the value of enjoying food while optimizing well-being [57].

### 4.2. Cultural Dimensions of Eating Styles

Eating behaviors and attitudes toward food are shaped by individual psychology and also by cultural norms, traditions, and community practices. Mindful and intuitive eating frameworks, largely developed in Western contexts, may require cultural adaptation to be meaningful and effective in diverse populations. For example, communal eating practices in Latin, African, and Asian cultures may center food as a tool for connection, healing, or storytelling, which are important values that can be harmonized with intuitive eating principles when framed appropriately. Additionally, acculturation stress among immigrants has been linked to disrupted eating patterns, including emotional eating and dietary displacement, where traditional diets are replaced by ultra-processed Western foods [64]. Recognizing and validating traditional foodways—such as fermented vegetables in Korean cuisine, legumes and tubers in Caribbean diets, or the use of medicinal herbs in South Asian cooking—can serve both as cultural anchors and therapeutic tools within nutritional psychiatry.

Recent cross-cultural research underscores the universality and adaptability of intuitive eating principles. For instance, Markey et al. [65] conducted a comprehensive survey across eight countries, revealing that intuitive eating is consistently associated with positive body image and self-esteem, regardless of cultural context. Similarly, Chammas et al. [66] examined intuitive eating among Lebanese adults, highlighting the role of cultural traditions and communal eating practices in shaping eating behaviors. These findings suggest that while the core principles of intuitive eating are broadly applicable, cultural nuances play a significant role in how these practices are adopted and experienced. Community-based interventions that incorporate culturally relevant food education, storytelling, and cooking practices show promise for fostering emotional regulation and dietary resilience while preserving identity. Adapting mindful eating curricula to reflect these values may improve engagement and outcomes in historically marginalized communities.

## 5. Dietary Patterns and Mental Health

While behavioral strategies such as mindful and intuitive eating promote internal self-regulation, the broader composition of the diet—its quality, diversity, and nutrient density—also plays a critical role in shaping mental health outcomes. A major challenge inherent in assessing the role of any dietary pattern with disease etiology (e.g., mental health disorders) is the complex range of demographic, biological, genetic, and behavioral determinants that contribute to disease development in any given population [67]. Dietary patterns that reduce the risk of mental health disorders are proposed to modulate neurological pathways associated with inflammatory processes, oxidative stress, brain plasticity, and the gut microbiome [67,68]. Therefore, the field of psychonutrition, though still emerging, has yielded promising evidence for the protective role of specific dietary patterns in mental health, particularly the Mediterranean diet and the DASH (Dietary Approaches to Stop Hypertension) diet [69]. The Mediterranean diet, characterized by high consumption of fruits, vegetables, whole grains, legumes, olive oil, and fish, is associated with improved mood, cognitive performance, and reduced risk of depression [30,70]. A systematic review by Brooks et al. [71] analyzed 23 studies and found that adherence to healthy dietary patterns, particularly Mediterranean and DASH diets, was associated with lower stress, anxiety, and depression scores across diverse populations. Diet may serve as a critical protective factor for mental health during childhood and adolescence. Similarly, Tan et al. [72] reviewed 16 studies involving 48,824 participants to examine the association between the DASH diet and mental well-being. Despite some inconsistencies in assessment methods across studies, they concluded that the DASH diet is negatively associated with depression symptoms and positively associated with mental health, quality of life, and emotional well-being [72].

Dietary quality appears particularly influential during sensitive developmental periods. In a recent systematic review of 13 studies on children and adolescents, Camprodon-Boadas et al. [70] highlighted the long-term cognitive and emotional benefits of early Mediterranean diet adoption as evidenced by a protective association with attention-deficit/hyperactivity disorder, depression, and anxiety. In a randomized controlled trial, Golmohammadi et al. [73] demonstrated that adherence to the Mediterranean–DASH Intervention for Neurodegenerative Delay (MIND) diet improved sleep quality and serum BDNF, a neurotrophin implicated in learning and emotional regulation, in women with overweight, diabetes, and insomnia. Meanwhile, the ketogenic diet has shown promise in psychiatric settings, particularly for treatment-resistant depression and bipolar disorder. Boltri et al. [74] outlined mechanisms involving the effect of ketone bodies on neurotransmitter balance, mitochondrial function, and anti-inflammatory signaling. Walaszek et al. [75] emphasized the potential of the ketogenic diet for relapse prevention when appropriately implemented under clinical supervision. However, despite some promising evidence in recent years, the ketogenic diet is associated with several areas of caution, including potential adverse effects (e.g., vitamin and mineral deficiencies, hepatic steatosis, hypoproteinemia, hypercalcemia, kidney stones, increased risk of heart disease and cognitive decline), the long-term health implications of which are still unknown [76].

Beyond specific dietary patterns, energy intake and nutrient timing have been examined for their effects on mood and cognition. For example, breakfast consumption appears to be a significant behavioral marker of mental health. Evidence from adolescent and adult populations suggests that habitual breakfast skipping is associated with elevated risks of depressive symptoms, irritability, and poor stress coping [77]. These associations may be partially mediated by glycemic instability and downstream effects on appetite and circadian rhythms [78]. Therefore, regular breakfast consumption may function as a simple, practical target within broader dietary interventions aimed at improving emotional resilience; however, time-restricted eating and other intermittent fasting approaches have demonstrated some potential positive effects. An editorial by Ammar et al. [69] synthesized evidence from eight studies, concluding that while diets like the Mediterranean, DASH, and MIND diets are linked to improved mood and reduced depressive symptoms, the effects of caloric restriction and intermittent fasting remain mixed and are highly context-dependent. These strategies improve metabolic flexibility and cognitive clarity in some individuals but could exacerbate disordered eating in others. Metabolic flexibility—i.e., the ability to efficiently switch between fuel sources—has been studied for decades [79,80] and offers a promising framework; it is still in early stages of clinical validation and applicability across populations [80]. Studies comparing whole foods versus isolated nutrients are also revealing. Importantly, Ali et al. [81] directly compared whole-food kiwifruit to matched vitamin C supplementation and found that only the whole fruit reduced cortisol and improved mood in women exposed to exercise-induced stress, pointing to the synergistic effects of whole-food matrices. This reinforces the idea that nutrient synergy within whole-food matrices may deliver more robust and sustainable mental health benefits than isolated nutrients.

Taken together, these findings underscore that nutrient combinations—rather than individual compounds—may offer the greatest potential for promoting mental health. While intermittent fasting and caloric restriction show promise for select outcomes, overall dietary quality remains a critical protective factor for physical and emotional well-being [82,83]. There is a critical need for continued research to clarify the roles of nutrient timing, gut–brain interactions, and individual variability in response to dietary interventions, reinforcing the role of nutrition as a meaningful adjunct in mental health care. While a range of dietary patterns, including the Mediterranean, DASH, and MIND diets, have been associated with improved mental health outcomes [67,84], the mechanisms underlying these benefits involve complex, interactive pathways. Nutrient density, glycemic stability, and anti-inflammatory effects all contribute to the modulation of mood and cognitive function [85]. However, these benefits are moderated by lifestyle, genetic, environmental, and psychosocial variables [86]. For example, despite the protective dietary profile of Mediterranean populations, mental health disparities persist [87], suggesting that socioeconomic status, access to care, social cohesion, and cultural practices also play critical roles. Rather than attributing outcomes to diet alone, it is important to consider diet as one factor, albeit a very important factor, in a broader biopsychosocial matrix.

## 6. Ultra-Processed Foods, Nutrition Inequality, and Public Health

Ultra-processed foods (UPFs) now make up over 50% of dietary intake in many Western countries and are consistently linked to adverse mental health outcomes [22,88]. UPFs are engineered for palatability and shelf stability but are stripped of fiber, micronutrients, and bioactive compounds, often containing additives that may disrupt the gut–brain axis. Lane et al. [22] conducted a meta-analysis and found that high UPF intake was associated with increased risk of depression across cohorts from North and South America, Europe, and Asia. In Brazil, Canhada et al. [88] confirmed that this relationship persisted across income and education subgroups, suggesting a widespread and structurally reinforced vulnerability.

The mechanisms underlying these associations are complex and evolving. While some consumer narratives describe UPFs, especially those high in sugar, as addictive, biological evidence is mixed. Rodda [89] documented how consumers describe themselves as “addicted” to sugar and classified 10 behavioral strategies to reduce sugar intake, most relying on emotional insight and environmental restructuring. However, PET scan data by Darcey et al. [90] did not show consistent dopamine responses to ultra-processed milkshakes (high fat, high sugar), challenging simplistic addiction models and questioning the purported mechanisms of action. Interestingly, post-ingestive dopamine responses to milkshakes were higher among participants reporting greater fasting hunger and subsequent, ad libitum cookie intake, indicating that motivational state and context may mediate reward signaling [90].

Beyond biology, nutrition literacy influences UPF consumption and related mental health risk. Mostafazadeh et al. [91] found that, even among nursing students, poor nutrition literacy correlated with higher stress and poorer eating behaviors. This finding underscores that the lack of food literacy extends even to health professionals, further complicating behavior-change efforts.

Inequity in food access exacerbates these patterns. Dunn et al. [17] observed that homeless youth with low omega-3 intake (especially EPA and DH) had significantly worse psychological scores than peers with dietary support, with particularly strong effects among females (inverse association between intake/erythrocyte status of omega-3 and depression). Similarly, Warren et al. [50] called attention to the compounding effects of toxic stress and poor diet on the microbiota–gut–brain axis, advocating for structural interventions and policy-level solutions. Cultural displacement further compounds these inequities. Traditional diets, often rich in whole foods and diverse plants, are frequently replaced by UPFs during migration or under socioeconomic pressure, leading to a loss of cultural identity and a rise in diet-related distress. Interventions that aim to restore or protect cultural food practices may support not only nutritional adequacy but also emotional well-being and resilience in the face of structural adversity. For example, among Native American communities, the revitalization of traditional foodways, such as gathering wild plants, growing heritage crops, and participating in communal food preparation, has been associated with improvements in diet quality, metabolic health, and mental well-being. Programs that reconnect individuals with ancestral knowledge and Indigenous food sovereignty not only address nutritional gaps but also restore cultural identity, self-determination, and intergenerational healing. This approach reframes dietary interventions as acts of resilience and reclamation, particularly powerful in populations historically subjected to food-related trauma and cultural erasure [92].

Collectively, these data demand public health systems to shift beyond individual responsibility and address the structural production, promotion, and accessibility of nutrient-poor foods. Nutrition interventions must be culturally grounded, equity-focused, and designed within a systems-thinking framework that reflects both biological vulnerability and social determinants of health.

## 7. Emerging Mechanisms

While the public health implications of poor diet quality are profound, emerging molecular and microbiome science offers insight into the biological mechanisms through which dietary exposures influence mental well-being.

### 7.1. Epigenetics, SCFAs, Metabolites

Nutrition influences not only metabolic health but also gene expression, inflammatory signaling, and neurotransmitter synthesis. For example, dietary methyl donors such as B vitamins, folate, and choline influence the DNA methylation of genes involved in neurodevelopment and stress resilience [1]. In addition to genetic factors, environmental factors (e.g., lifestyle factors including diet, physical activity, stress, coping mechanisms, early life experiences, socioeconomic status) underscore the relationship between physiology and mental well-being [1]. Nutrients derived from diet and supplementation affect gut microbiota and epigenetic mechanisms [93]. Maintaining adequate levels of important nutrients (e.g., vitamin D, iron, fiber, zinc, magnesium) may decrease inflammation via the relationship between the microbiota and epigenetic modulation [93], which also carries implications for mental well-being.

Food components that exert beneficial epigenetic effects on mental well-being are being explored, especially in the context of intestinal barrier protection by way of microbiome regulation. Lutz et al. [53] proposed that genetic predispositions for mood disorders may be expressed or silenced depending on the dietary intake of omega-3s, fiber, and polyphenols. This opens new frontiers in “nutriepigenomics,” where diet modulates risk at the transcriptional level. Further, Raza et al. [94] explored how vitamin D and omega-3 fatty acids regulate inflammatory pathways and monoamine function, both essential for mood stabilization. Citrus intake may have specific microbial effects. Samuthpongtorn et al. [55] reported that *F. prausnitzii*, which produces anti-inflammatory metabolites, mediated the association between citrus intake and depression symptoms in a large prospective cohort. A systematic review by Gabriel et al. [95] examined the nutritional links in bipolar disorder, identifying magnesium, vitamin B6, and omega-3 fatty acids as recurrent deficiencies in affected individuals. Taken together, the molecular effects of nutrients on gene regulation, inflammatory signaling, and microbial activity underscore their far-reaching influence on brain health.

Dietary protein plays a key role in mental health by supplying essential amino acids (e.g., tryptophan, phenylalanine, and tyrosine) that are critical for neurotransmitter synthesis, including serotonin and dopamine [96]. These neurotransmitters regulate mood, cognition, and emotional stability, and imbalances have been linked to various mental disorders [96]. Plant-based proteins offer numerous health and environmental effects but may have less bioavailability, which is an important consideration in populations who may require higher amounts (e.g., aging populations) [97]. Notably, while protein can be derived from both plant and animal sources, recent evidence suggests plant proteins are not associated with mental disorders, but a diet high in animal protein may predispose individuals to mental illness [96].

These findings collectively support a paradigm shift: nutrients act not only as sources of energy or structural components but also as signaling molecules that influence gene expression, inflammation, neurotransmission, and microbiome activity. Through these complex metabolic and epigenetic pathways, the brain dynamically responds to dietary patterns—reinforcing the potential of targeted nutrition to modulate mental health at a molecular level.

### 7.2. Precision Nutrition as a Mental Health Intervention

Building on these mechanistic insights, precision nutrition offers a promising frontier in mental health by tailoring dietary interventions based on an individual’s unique genetic, epigenetic, microbiome, and metabolic profiles. In this model, interventions go beyond general dietary guidelines to account for variability in nutrient metabolism, microbiota composition, and inflammation pathways that influence mood and cognition [98,99]. For example, individuals with specific polymorphisms in genes such as MTHFR or BDNF may respond differently to folate or omega-3 supplementation, respectively [32,53]. Similarly, microbiome-informed diets can target microbial taxa associated with depressive symptoms, such as a low abundance of *Faecalibacterium prausnitzii* or elevated pro-inflammatory strains [54,55]. For example, a patient with low *F. prausnitzii* abundance and an MTHFR polymorphism might benefit from a Mediterranean-style diet rich in citrus, leafy greens, and fermented foods, alongside methylated folate supplementation—targeting both microbial diversity and nutrient metabolism.

Despite its promise, precision nutrition faces several limitations, including cost, lack of access to validated tools in clinical settings, and insufficient longitudinal data to predict individual response. As multi-omic technologies advance and become more accessible, personalized dietary strategies are likely to emerge as a powerful complement to conventional therapies in nutritional psychiatry—offering the potential for more targeted, effective, and sustainable interventions. This is especially the case for those who are less responsive to traditional dietary recommendations due to differing biology or cultural, behavioral, or psychological reasons.

## 8. Implementation and Policy Gaps

Despite strong empirical support, the translation of nutritional psychiatry into policy remains slow. Su et al. [100] highlighted the gap between research and implementation and called for embedding nutrition into primary care protocols for depression and anxiety.

A national framework for action was outlined by Higgs et al. [101] that reinforces the importance of improving access to minimally processed foods, regulating UPF marketing, integrating nutrition into mental health training, and funding large-scale dietary intervention trials. With the importance of mental well-being in mind, Montgomery et al. [11] conducted a meta-analysis and concluded that dietary interventions yield modest but reliable improvements in mental health outcomes, often exceeding those of standard psychoeducation in vulnerable populations.

Public health strategies aimed at healthy aging and chronic disease prevention also align with mental health promotion. Tessier et al. [102] argued that dietary patterns optimized for healthy aging (e.g., Mediterranean diet, MIND) should be central to public health strategy, while Toledo [103] pointed to the important contribution of vegetable intake as a modifiable risk factor for major depression. Similarly, Schweren [104] demonstrated that diet quality remained an independent predictor of mental health outcomes in a cohort of over 120,000 adults, controlling for demographic and lifestyle factors.

These findings support that implementation must move beyond individual responsibility and become a coordinated effort across healthcare, education, agriculture, and public policy.

### 8.1. Clinical Implications

Nutrition plays an indispensable role in both physical and mental health outcomes, yet its integration into primary healthcare remains limited. Chao et al. [105] highlight significant gaps in the inclusion of nutrition education within primary care adult and family nurse practitioner (ANP/FNP) programs. Their cross-sectional survey of faculty across U.S. programs revealed that students received an average of only 14.4 h of nutrition education, with only a quarter of institutions meeting the recommended 25 h benchmark. Faculty reported perceiving nutrition education as critically important, but systemic barriers such as curricular overcrowding and a lack of qualified faculty persist. These deficits in provider education have downstream effects, limiting the provision of evidence-based dietary counseling, which is foundational to chronic disease prevention and management, including conditions closely associated with mental health, such as obesity, diabetes, and cardiovascular diseases [105].

The World Health Organization’s policy brief, as synthesized by Kraef et al. [106], underscores the economic and systemic burden of malnutrition, noting its contribution to one in five deaths globally and its substantial strain on health systems, particularly in low- and middle-income countries. The authors advocate for comprehensive primary healthcare as a critical platform for addressing the double burden of undernutrition and obesity, positioning nutrition as a human rights issue intertwined with universal health coverage (UHC). Importantly, they argue that primary healthcare settings offer unique opportunities to integrate promotive, preventive, curative, and rehabilitative nutrition interventions that also address the social and commercial determinants of health. Their recommendations include fostering community empowerment and promoting policies that prioritize equitable access to nutrition services, which has direct implications for mental health well-being by ensuring that vulnerable populations receive culturally appropriate and holistic care [106].

Given these relationships, clinical implementation should prioritize dietary patterns shown to reduce inflammation and support mental health (e.g., Mediterranean, MIND, or anti-inflammatory whole-food diets) but do so through tailored, collaborative strategies that respect patient context and readiness. Emphasis should shift from what to eat toward how clinicians can co-create nutrition plans that feel sustainable and empowering. Collaborative nutrition plans should utilize evidence-based recommendations for the use of specific nutraceuticals (e.g., omega-3 fatty acids, vitamin D, probiotics, and selected herbal remedies) as adjunctive interventions for mood and anxiety disorders, bipolar disorder, and schizophrenia, supported by guidelines from international psychiatric bodies [107].

Beyond dietary recommendations, Fenton et al. [107] advocate for an interdisciplinary, person-centered approach that integrates nutrition into routine mental health care. They emphasize the role of healthcare providers in assessing dietary patterns, nutritional deficiencies, and related lifestyle factors within comprehensive psychiatric evaluations, while collaborating with nutrition professionals to support individualized interventions. The authors further call for the tailoring of nutritional strategies to patients’ psychosocial contexts, ensuring that care is equitable, culturally sensitive, and responsive to individual preferences and barriers [107]. This integrative model not only addresses the nutritional determinants of mental health but also promotes overall well-being, resilience, and chronic disease prevention, reinforcing the imperative to embed nutrition as a core element of holistic mental health practice.

Collectively, these works point to a pressing need to enhance nutrition literacy among healthcare providers, integrate nutrition into primary care settings, and implement community-based interventions that address the psychosocial dimensions of dietary behaviors. These measures are essential for advancing population mental health and well-being, aligning with both clinical and public health priorities. Clinicians have an opportunity to screen for nutrition-related behaviors, including meal regularity, dietary diversity, and emotional eating patterns, as part of routine assessments in patients with stress-related disorders [108,109]. Personalized nutrition counseling that incorporates interoceptive awareness, flexible meal planning [110], and trauma-informed care may enhance both treatment engagement and mental health outcomes [111]. Additionally, interdisciplinary collaboration between dietitians, psychologists, and primary care providers is critical to bridge the gap between metabolic and psychological health [112,113]. Recognizing the bidirectional relationship between diet and emotional regulation [114], clinical settings may benefit from integrating brief behavioral nutrition interventions that are health-centric to empower patients through self-regulation and body trust, rather than solely weight-focused goals [57].

### 8.2. Interprofessional Collaboration

Integrating nutrition professionals such as Registered Dietitian Nutritionists (RDNs) and Certified Nutrition Specialists (CNSs), alongside behavioral health providers, into collaborative care teams enhances the management of stress-related eating behaviors and mental health conditions. The Collaborative Care Model (CoCM), an evidence-based framework, facilitates this integration by promoting systematic coordination among primary care providers, care managers, and psychiatric consultants [113]. This model has demonstrated efficacy in improving outcomes for depression and anxiety, particularly in diverse and underserved populations [113,115]. Similarly, the Veterans Health Administration’s Whole Health approach exemplifies a comprehensive, team-based strategy that incorporates nutrition, mental health, and complementary therapies to address the holistic needs of patients [116]. Despite these advancements, challenges such as limited cross-disciplinary training, reimbursement complexities, and unclear role delineations persist. Addressing these barriers through enhanced education, practice protocols, and supportive policies can foster effective interprofessional collaboration, ultimately improving outcomes in nutritional psychiatry.

## 9. Conclusions

Emotional eating is not simply an issue of willpower; it is a neurobiological, psychological, and environmental phenomenon reinforced by chronic stress and poor food quality. However, the science is also hopeful: food can regulate mood, build emotional resilience, and interrupt maladaptive behavior patterns when used strategically. Moreover, food is not merely a source of macro- and micronutrients, but also a medium for social connection, cultural identity, and enjoyment. Mindful and intuitive eating re-engage the body’s internal regulation systems while diets rich in phytochemicals, fiber, and healthy fats nourish the gut–brain axis and reduce neuroinflammation. The avoidance of ultra-processed foods and improving equitable access to whole, nutrient-dense foods may represent the most sustainable strategy to promote well-being.

Integrating culturally relevant dietary approaches and honoring traditional foodways can further support the restoration of a healthy relationship with food, particularly in diverse and underserved communities. As we move into an era of personalized and preventive care, it is essential to embed nutrition not only as a determinant of physical health but as a foundation for emotional regulation, cognitive resilience, and, ultimately, whole-person well-being.

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
