# Peer review of "Restoring a Healthy Relationship with Food by Decoupling Stress and Eating: A Translational Review of Nutrition and Mental Health"

_nutrients, 2025, doi:10.3390/nu17152466_

Round 1

Reviewer 1 Report

Comments and Suggestions for Authors

This narrative review provides important information and messages that underscore the role of healthy diet and nutrition in the prevention and treatment of mental diseases. However, the overall impression is that it lacks focus. On the other hand, the authors do not provide adequate information on each item.

Comments

  1. In Abstract, the authors write “the paper highlight blunted interoception”. However, we could not find any detailed description about that.
  2. The authors seem to place an emphasis on gut microbiota. However, little is written about how individual organisms affect body systems. I feel the authors should refer to roles of important beneficial microbes such as Bifidobacterium and Lactobacillus.
  3. “Trauma-informed approaches” are not fully explained.
  4. In the 5. Dietary Patterns and Mental Health section, skipping breakfast (perhaps one of the most important factors) is not referred to.
  5. There are some repetitions of contents. For example, the content of the 3rd paragraph of 8.1.Clinical implications is already written in 5. Dietaty Patterns and Mental Health section.
  6. Many references are incomplete; ref 4, 6, 9, 11, 13, etc. have not contain journal titles, volumes and pages.

Author Response

Reviewer 1:

Comments and Suggestions for Authors

This narrative review provides important information and messages that underscore the role of healthy diet and nutrition in the prevention and treatment of mental diseases. However, the overall impression is that it lacks focus. On the other hand, the authors do not provide adequate information on each item.

  • Thank you for this valuable feedback. We value this assessment and insight and have made edits to our manuscript accordingly. We are grateful for these improvements to our manuscript. Please see details as follows:

Comments

  1. In Abstract, the authors write “the paper highlight blunted interoception”. However, we could not find any detailed description about that.
    • Thank you for pointing this out, we expanded on this concept in section 2 (Stress, emotion, eating behavior) at the end of the first paragraph, as well as the last paragraph in 4.1
  2. The authors seem to place an emphasis on gut microbiota. However, little is written about how individual organisms affect body systems. I feel the authors should refer to roles of important beneficial microbes such as Bifidobacterium and Lactobacillus.
    • Thank you for this suggestion – this could actually be a paper unto itself so we tried not to dive too far into the weeds in this section, but we appreciate your idea and we have added this to the last paragraph under section 3 (Neurobiology, gut-brain axis, and inflammation).
  3. “Trauma-informed approaches” are not fully explained.
    • Thank you – we have expanded this under section 2 (stress, emotion, and eating behavior) in the second to last paragraph.
  4. In the 5. Dietary Patterns and Mental Health section, skipping breakfast (perhaps one of the most important factors) is not referred to.
    • Thank you – we have added this to the 3rd paragraph in section 5.
  5. There are some repetitions of contents. For example, the content of the 3rd paragraph of 8.1.Clinical implications is already written in 5. Dietaty Patterns and Mental Health section.
    • We have removed a paragraph under 8.1 to keep the focus more on implementation of clinical applications.
  6. Many references are incomplete; ref 4, 6, 9, 11, 13, etc. have not contain journal titles, volumes and pages.
    • Thank you for catching this – this was an error with our citation software and our oversight. Please see the corrected references.

Reviewer 2 Report

Comments and Suggestions for Authors

The review from Warren and Frame titled “Restoring a Healthy Relationship with Food by Decoupling Stress and Eating: A Translational Review of Nutrition and Mental Health” aimed to synthesize interdisciplinary findings from nutritional psychiatry, microbiome science, and behavioral nutrition to explore how stress physiology, gut-brain interactions, and dietary quality shape emotional regulation and eating behavior. It highlights mechanisms (e.g. HPA-axis dysregulation, blunted interoception, and inflammatory and epigenetic pathways) and examines the evidence for mindful and intuitive eating; hytochemical-rich, whole-food dietary patterns; and the emerging role of precision nutrition.

General comment

The paper is well written and the salient literature has been well highlighted by the authors. The topic is really important and current and the information reported can be very useful for future studies. In the last two decades, nutrition and eating behavior are emerging as an important factor not only in determining some imbalances, but also as a therapy.

Major

Paragraph 2. Stress, Emotion, and Eating Behavior.

1) “At its core, stress activates the HPA axis, resulting in elevated cortisol secretion, which in turn increases appetite, especially for energy-dense, high-sugar, and high-fat foods”.

I’ don’t completely agree because this is not true for all individuals. In fact, it is still unclear why some individuals under stress eat a lot and gain weight, while others refuse food and lose weight. Furthermore, even in the same individual, during the course of one's life, the situation can change. What are the possible causes?

2) “…increased snack frequency and poorer nutritional choices across age and gender groups”.

I think that the differences related to age and gender should be better explained.

3) What do the authors mean by "gender"?

Paragraph 3. Neurobiology, Gut-Brain Axis, and Inflammation

1) “In contrast, short chain fatty acids (SCFAs), primarily butyrate, acetate, and propionate, produced through microbial fermentation of dietary fiber in the gut, exhibit anti-inflammatory effects and influence brain signaling via the vagus nerve and blood-brain barrier, e.g. permeability.”

This is an interesting concept, because normally short-chain AGs are harbingers of disorders. I think it deserves more attention.

Paragraph 5. Dietary Patterns and Mental Health

1) In this paragraph, the authors limit themselves to a list of what has been found, said, highlighted by others, but they do not put together the information and argue it to help readers draw usable conclusions or considerations.

“The Mediterranean diet, characterized by high consumption of fruits, vegetables, whole grains, legumes, olive oil, and fish, is associated with improved mood, cognitive performance, and reduced risk of depression (49,50).”

2) However, it is not that depressive conditions do not exist in Mediterranean populations; so, what other factors should be considered that can counteract the benefits of the Mediterranean diet?

3) “potential of ketogenic diet”.

The ketogenic diet and bariatric surgery are still too recent to have clear information on their long-term effects. These aspects should be discussed

4) “These strategies by improve metabolic flexibility and cognitive clarity in some individuals but could exacerbate disordered eating in others.”

Metabolic flexibility is also a very recent field.

Paragraph 7. Emerging Mechanisms

1) What is the role of proteins instead? Especially those with high biological value?

Author Response

The review from Warren and Frame titled “Restoring a Healthy Relationship with Food by Decoupling Stress and Eating: A Translational Review of Nutrition and Mental Health” aimed to synthesize interdisciplinary findings from nutritional psychiatry, microbiome science, and behavioral nutrition to explore how stress physiology, gut-brain interactions, and dietary quality shape emotional regulation and eating behavior. It highlights mechanisms (e.g. HPA-axis dysregulation, blunted interoception, and inflammatory and epigenetic pathways) and examines the evidence for mindful and intuitive eating; hytochemical-rich, whole-food dietary patterns; and the emerging role of precision nutrition.

General comment

The paper is well written and the salient literature has been well highlighted by the authors. The topic is really important and current and the information reported can be very useful for future studies. In the last two decades, nutrition and eating behavior are emerging as an important factor not only in determining some imbalances, but also as a therapy.

  • Thank you for taking the time to provide this valuable feedback. We truly believe it has improved the quality of our manuscript. Please see our responses below and associated edits in the manuscript.

Major

Paragraph 2. Stress, Emotion, and Eating Behavior.

1) “At its core, stress activates the HPA axis, resulting in elevated cortisol secretion, which in turn increases appetite, especially for energy-dense, high-sugar, and high-fat foods”.

I’ don’t completely agree because this is not true for all individuals. In fact, it is still unclear why some individuals under stress eat a lot and gain weight, while others refuse food and lose weight. Furthermore, even in the same individual, during the course of one's life, the situation can change. What are the possible causes?

  • Thank you – this is an important point. Although generally most people respond with increased appetite and drive for highly palatable foods, some have the opposite response. Furthermore, the longer the stress response endures, high cortisol levels may ultimately become a blunted cortisol response with lower levels which may result in further variation. We have added information to highlight these important variations.

2) “…increased snack frequency and poorer nutritional choices across age and gender groups”.

I think that the differences related to age and gender should be better explained.

  • Thank you for this suggestion – we have expanded this discussion under section 2 (stress, emotion, and eating behavior)

3) What do the authors mean by "gender"?

  • This is an important distinction – we have conceptualized this in the last paragraph of the introduction.

Paragraph 3. Neurobiology, Gut-Brain Axis, and Inflammation

1) “In contrast, short chain fatty acids (SCFAs), primarily butyrate, acetate, and propionate, produced through microbial fermentation of dietary fiber in the gut, exhibit anti-inflammatory effects and influence brain signaling via the vagus nerve and blood-brain barrier, e.g. permeability.”

This is an interesting concept, because normally short-chain AGs are harbingers of disorders. I think it deserves more attention.

  • This is definitely an interesting concept – thank you for allowing the space to expand this. We have added text to this paragraph.

Paragraph 5. Dietary Patterns and Mental Health

1) In this paragraph, the authors limit themselves to a list of what has been found, said, highlighted by others, but they do not put together the information and argue it to help readers draw usable conclusions or considerations.

  • We understand your point and have added text to help the reader draw usable conclusions and considerations.

“The Mediterranean diet, characterized by high consumption of fruits, vegetables, whole grains, legumes, olive oil, and fish, is associated with improved mood, cognitive performance, and reduced risk of depression (49,50).”

2) However, it is not that depressive conditions do not exist in Mediterranean populations; so, what other factors should be considered that can counteract the benefits of the Mediterranean diet?

  • Thank you for this prompting question – we have added text to include these factors throughout section 5.

3) “potential of ketogenic diet”.

The ketogenic diet and bariatric surgery are still too recent to have clear information on their long-term effects. These aspects should be discussed

  • We agree with this wholeheartedly and have added the adverse effects and unknown long-term health implications of the ketogenic diet. We prefer to leave bariatric surgery (and other recent interventions such as GLP-1 agonists) out of the discussion to maintain focus on dietary patterns.

4) “These strategies by improve metabolic flexibility and cognitive clarity in some individuals but could exacerbate disordered eating in others.”

Metabolic flexibility is also a very recent field.

  • Thank you for pointing this out – while it has been studied for a few decades it is just now emerging as a clinical consideration and therapeutic target – we have added text to highlight this.

Paragraph 7. Emerging Mechanisms

1) What is the role of proteins instead? Especially those with high biological value?

  • Excellent point – we have added this under the Emerging Mechanisms section.

Reviewer 3 Report

Comments and Suggestions for Authors

Warren and Frame provide a manuscript entitled: “Restoring a healthy relationship with food by decoupling stress and eating: a translational review of nutrition and mental health. The manuscript is interesting”. However, some points need to be addressed.

  • The introduction is very short and must be extended providing more information. For example the Author must extended in general the discussion about diet and mental heath by including also beverages besides food. Beverages including coffee may have positive effects on perceived stress (DOI: 10.3390/antiox12020272). The Authors may consider to add these and other findings in the manuscript, which should support the objectives of this review

  • The authors must also extend the part clinical implications because this is a very important aspect.

  • The Authors should also briefly discuss sex/gender differences in this context. stress-related disorders are more common on women than in men.

  • The Authors must check for typos throughout the manuscript.

  • The Authors must also check for the presence of statements without references throughout the manuscript.

Author Response

Warren and Frame provide a manuscript entitled: “Restoring a healthy relationship with food by decoupling stress and eating: a translational review of nutrition and mental health. The manuscript is interesting”. However, some points need to be addressed.

  • Thank you for your insightful feedback. We agree with your suggestions and have incorporated these edits into our manuscript, which we believe has improved its quality greatly. Please see our comments below and associated edits in the manuscript.

  • The introduction is very short and must be extended providing more information. For example the Author must extended in general the discussion about diet and mental heath by including also beverages besides food. Beverages including coffee may have positive effects on perceived stress (DOI: 10.3390/antiox12020272). The Authors may consider to add these and other findings in the manuscript, which should support the objectives of this review
    • Thank you for bringing this up – beverages are such an important part in the context of dietary patterns as a whole, and thank you for the reference. While we had assumed them under diet, we have clarified this in our introduction.

  • The authors must also extend the part clinical implications because this is a very important aspect.
    • Thank you for this suggestion – we agree and have expanded this section

  • The Authors should also briefly discuss sex/gender differences in this context. stress-related disorders are more common on women than in men.
    • This is an excellent point – we have added this to our introduction.

  • The Authors must check for typos throughout the manuscript.
    • Thank you, we have double-checked the manuscript and fixed typos

  • The Authors must also check for the presence of statements without references throughout the manuscript.
    • Thank you, we have double-checked this and added references where appropriate.

Round 2

Reviewer 1 Report

Comments and Suggestions for Authors

I feel the manuscript has improved substantially.

Reviewer 2 Report

Comments and Suggestions for Authors

I commend the authors for having greatly improved their manuscript.

Reviewer 3 Report

Comments and Suggestions for Authors

The Authors have successfully addressed all the points I raised.